# Vision Measurement System for Gender-Based Counting of *Acheta domesticus*

**DOI:** 10.3390/s24154936

**Published:** 2024-07-30

**Authors:** Nicola Giulietti, Paolo Castellini, Cristina Truzzi, Behixhe Ajdini, Milena Martarelli

**Affiliations:** 1Dipartimento di Ingegneria Industriale e dell’Informazione, Università degli Studi di Pavia, Via Adolfo Ferrata 5, 27100 Pavia, Italy; 2Department of Industrial Engineering and Mathematical Science, Università Politecnica delle Marche, Via Brecce Bianche 12, 60131 Ancona, Italy; p.castellini@staff.univpm.it (P.C.); m.martarelli@staff.univpm.it (M.M.); 3Department of Life and Environmental Science, Università Politecnica delle Marche, Via Brecce Bianche 12, 60131 Ancona, Italy; c.truzzi@staff.univpm.it (C.T.); b.ajdini@pm.univpm.it (B.A.)

**Keywords:** vision-based measurement systems, clustering techniques, image processing, measurement uncertainty assessment

## Abstract

The exploitation of insects as protein sources in the food industry has had a strong impact in recent decades for many reasons. The emphasis for this phenomenon has its primary basis on sustainability and also to the nutritional value provided. The gender of the insects, specifically *Acheta domesticus*, is strictly related to their nutritional value and therefore the availability of an automatic system capable of counting the number of Acheta in an insect farm based on their gender will have a strong impact on the sustainability of the farm itself. This paper presents a non-contact measurement system designed for gender counting and recognition in *Acheta domesticus* farms. A specific test bench was designed and realized to force the crickets to travel inside a transparent duct, across which they were framed by means of a high-resolution camera able to capture the ovipositor, the distinction element between male and female. All possible sources of uncertainty affecting the identification and counting of individuals were considered, and methods to mitigate their effect were described. The proposed method, which achieves 2.6 percent error in counting and 8.6 percent error in gender estimation, can be of significant impact in the sustainable food industry.

## 1. Introduction

The house cricket, *Acheta domesticus* (AD), recently authorized by the European Union as novel food (EU 2022/188) is one of the most predominant insect species widely consumed by humans [1], due to their high nutritional value, with a high protein content and a good index of essential amino acids compared to traditional protein sources [2,3,4,5,6,7]. The nutritional value of insects depends on their feed [8,9], developmental stage [10], rearing parameters (e.g., temperature, humidity) and insect processing [11]. Some studies indicate that gender plays a significant role in determining the nutritional value of insects [12]. Kulma et al. [13] demonstrated that females of AD contain a significantly higher amount of lipids and fewer proteins than males. Moreover, males contain more chitin and nitrogen chains than females. In recent years, the significant increase in AD consumption as a food source in Europe underscores the need to study the impact of AD gender on its nutritional profile. Currently, few studies exist due to the lack of technology for automatically distinguishing between male and female crickets, making manual separation time-consuming. The male-to-female ratio in farmed house crickets is crucial for balancing reproduction and production. Optimizing rearing procedures is essential to maximize reproductive yield and meet the growing market demand for ADs as food. In optimal conditions, the male-to-female ratio within a cricket farm tends to be balanced with a proportion of 1:1 [14]. This ratio ensures a uniform distribution of fertility and stable egg production. A flexible approach based on the specific needs of the farm should allow to optimize production and to respond effectively to market demands. Currently, there are no effective technologies capable of determining the relationship between males and females in a cricket farm, except via human visual check, but this is time-consuming, prone to error, operator dependent and not an economically viable route for cricket rearing farms. Hence, there is a need to create an automatic system for counting male and female specimens in cricket farms, which would allow to balance the male–female ratio. This contributes not only to the process optimization in insect mass rearing but also to the quality and quantity of insects produced, with a view to greater yield in breeding and a better knowledge of the nutritional characteristics related to the gender.

### 1.1. Motivation

In order to overcome the limitations associated with manual counting, in this paper we want to present the design and characterization of an automatic measurement system for gender-based counting of ADs in real-case scenarios. Counting and dividing the crickets by gender within the cricket farms would be difficult as they would be occluded from view, overlapping and in the presence of heavy dirt. To solve this problem, an identical cricket farm is proposed to be used where the entire cricket colony can be transferred. The two farms are connected by a transparent tube. The crickets spontaneously move from one farm to the other because the new one is made more attractive (e.g., through modulation of lights, water, food, temperature, etc.). The proposed system observes ADs, stimulated to pass through the transparent tube, which locates and distinguishes exploiting a Deep-Learning-based semantic segmentation model. The system takes advantage of the fact that female crickets are easily recognized by the ovipositor, an external organ with which the females of some animals, including ADs, are equipped to lay eggs into soil. The semantic segmentation model identifies all those pixel correspondent the position of the cricket within the image and assigns them a class based on gender. The data acquired are analyzed to count ADs and thus enable estimations of the size of the cricket population, with specific emphasis on the percentage of females. This analysis implements several selection criteria to reduce the effects of various disturbances and outliers. The development of the counting procedure is followed by a comprehensive uncertainty analysis of both the semantic segmentation model and the post-processing algorithm for gender-based counting.

### 1.2. Computer Vision and Artificial Intelligence for Agriculture and Entomology

In recent years, image processing with Artificial Intelligence (AI) models is widely used in different areas of application [15]. In more detail, if we consider food, agriculture, entomology and breeding of edible insects industry discovered very promising applications [16,17]. In [18], a quite comprehensive review of current studies about stored grains is given. Internet of Things (IoT) and Machine Learning (ML) technologies are applied to monitor the quality of stored grains, in real-time. In [19], an expert system is described for measuring and recognizing the quality and purity of mixed raisins. A machine vision setup made possible to capture images and process them with gray-level histograms, Gray Level Co-occurrence Matrix (GLCM), Gray Level Run-length Matrix (GLRM), and Local Binary Pattern (LBP). Principal Components Analysis (PCA) made possible the selection of extracted features by means of Artificial Neural Network (ANN) and Support Vector Machine (SVM) were used for classifying them. In [20], an integrated insects monitoring procedure that employs a vision system and data-driven Deep-Learning models is proposed. Six common insect species (i.e., *Lasioderma serricorne*, *Stegobium paniceum*, *Tribolium castaneum*, *Sitophilus oryzae*, *Oryzaephilus surinamensis* and *Trogoderma variabile Bailon*) can be detected and identified in warehouses food facilities, and retail environments. The images are processed using YOLO model in different variants (YOLOv5 and YOLOv8), providing an end-to-end framework for automatic and real-time insect detection and classification in a real environment. In [21], Integrated Pest Management (IPM) is supported by an automatic detection system. It is based on a camera, a micro-computer, a deep learning model based on backbone DPeNet. The system is compared with Faster R-CNN, SSD, and YOLOv3 demonstrating its capabilities. In [22], an insect pest recognition system was developed based on a feature fusion network to synthesize feature presentations in different backbone models. The system employ one CNN-based backbone ResNet, and two attention-based backbones Vision Transformer and Swin Transformer to localize the insect images, with particular attention to the robustness to augmented images. In [23], the leading techniques for the automated detection of insects are outlined, highlighting the most successful approaches and methodologies. In [24], AI-based approaches are analysed for a wide spectrum of agricultural problems, the most challenging being pest management. The proposed method was based on Convolutional Neural Network (CNN), enabling to classify in real-time two tephritid species (*Ceratitis capitata* and *Bactrocera oleae*) in real environment. In [25], a vision system, capable of detecting, tracking, and identifying individual insects in natural environment, is presented: it is based on deep learning open source software.

### 1.3. Novel Contribution

The main limitation of previous studies applied to insects detection and classification tasks (see Section 1.2) is that no solutions have been presented for identifying gender or measuring gender ratio. More importantly, the metrological performance of these approaches has not been analyzed, leading to various sources of uncertainty. Montgomery et al. in [26] discuss the need for standardized methods in insect monitoring to address the lack of consistent data and metrological performance in previous studies and emphasize the importance of standardized data collection practices to improve the accuracy and integration of insect population data on a global scale. Several studies in recent years have shown promising results in related areas, providing a basis for current research [25,27]. In [28], Zacares et al. explore the use of computer vision techniques for sex selection of various mosquito species based on pupal size dimorphism. The results contribute to the development of mass mosquito breeding and sterile insect technique (SIT) programs. As can be deduced from the study of the state-of-the-art presented in Section 1.2, the main limitation of the previous studies applied to specimen recognition is that no solutions have been presented to identify gender or to measure gender ratio. In summary, the innovations presented in this research works are:to design and realize a test bench, which can be easily implemented in a farming house, allowing to observe insects one by one while migrating from an environment to another;to implement a gender based recognition method using machine vision;to develop a counting methodology taking into account the disturbing sources and limiting their effect on the female/male ratio calculation;to perform a comprehensive uncertainty analysis of the whole procedure consisting of insect recognition and gender counting.

### 1.4. Paper Structure

The remainder of the paper is organised as follows: The experimental setup, involved data-driven semantic segmentation model and output data organisation are described in Section 2; the post-processing procedure for ADs tracking and counting is presented in Section 3; the results are discussed in Section 4, where the uncertainty sources are also presented. Finally, Section 5 draws the conclusion and reports future development.

## 2. Material and Methods

### 2.1. Experimental Set-Up

The test bench consists of two equal plastic containers (BOX1 and BOX2) with a size of 70 × 50 × 40 h cm^3^, named BOX1 and BOX2, see Figure 1. ADs are deposited in BOX1, which is kept lighted and devoid of food and water, and are attracted to move toward BOX2, which is kept in darkness and provided with food and water. The two boxes are connected by a passageway (5 × 3 × 32 cm^3^) made of transparent perspex underneath which the measurement camera and its lighting system are installed. A camera of 1920 × 1080 px resolution is connected to an acquisition computer and made to operate at a rate of 5 fps. The duration of the acquisition was 35.4 h (i.e., time that was taken by the whole AD colony to move from BOX1 to BOX2) and therefore 457,000 frames have been acquired. At the beginning, a certain number of ADs has been introduced in BOX1 for a total weight of 1100 g. By performing a manual count of the ADs present in BOX2 at the end of the test, it has been estimated 534 males and 747 females, for a total of 1281 insects.

### 2.2. AI-Based AD Detection and Gender Classification

The architecture of YOLOv8 for segmentation tasks, developed in [29], is chosen as the semantic segmentation model. YOLOv8, released in 2023, has demonstrated success in several real-time object detection and semantic segmentation applications [30]. Automating the detection and classification of ADs poses special challenges, especially due to the absence of a publicly available dataset. This section outlines the comprehensive methodology used to create a customized dataset and optimize hyper-parameters for training a YOLOv8 segmentation model. YOLOv8 architecture is chosen for its accuracy and low computational demands, that introduces significant improvements in the ability to detect small objects and handle complex backgrounds, making it ideal for the application that is the subject of this paper. Only a portion of the total collected frames is involved in training. Therefore, 20,000 frames are randomly selected from the total 457,000. Through Python software developed specifically for the application, the collected frames are manually labelled by an experienced operator. A two-class semantic segmentation dataset is created for male and female ADs. To generate this dataset, the expert operator manually selects pixel belonging to ADs and enters their class based on their or her experience. Once the dataset is collected, a custom training of the YOLOv8 architecture is performed. The Mean Average Precision (mAP) at a 50–95% Intersection over Union (IoU) threshold (mAP@.50-95), a common metric for assessing object detection and semantic segmentation models [31], is chosen as metric to be optimized for the semantic segmentation task. It combines precision and recall and provides a single measure of quality across all detection thresholds. The formula for mAP@.50-95 is given by:(1)$$\mathrm{mAP@}.50\text{-}95=\frac{1}{N}\sum_{i=1}^{N}{\mathrm{AP}}_{i}\mathrm{@IoU}=0.50$$
where *N* is the number of classes (e.g., male and female), and ${\mathrm{AP}}_{i}$ is the Average Precision for class *i* at an IoU threshold of 0.50. The hyper-parameters for model training are selected with a Bayesian optimization technique exploiting the Python implementation in Optuna library [32]. This technique, which has been used in several fields, has been shown to be superior to others in selecting hyper-parameters even in complex situations [33,34]. Once the objective function to be optimized is chosen (i.e., mAP@.50-95), the values of the variables (i.e., hyperparameters) are continuously updated (i.e., repeated training with different hyper-parameters are performed) based on the results of previous iterations. In this way, the space of variables is explored and the optimal configuration is converged to maximize the objective function. In this specific case, the objective function to be maximized is mAP@.50-95. The training epochs are fixed at 1000, with an early stopping mechanism set to 50 on the validation metrics (i.e., mean mAP@.50-95 calculated in validation dataset). The initial learning rate value is varied between 0.001 and 0.000001, the batch size between 2 and 32, the optimizer for training chosen from a set of available optimizers (i.e., SGD, Adam, AdamW, NAdama, RAdam, and RMSProp [35]), and the architecture chosen from various configurations of YOLOv8 (i.e., YOLOv8n-seg, YOLOv8s-seg, YOLOv8m-seg, YOLOv8l-seg, and YOLOv8x-seg) [29]. The input frame size is fixed at 640 × 640 px. Each individual training is cross-validated by k-fold method with 3 folds. The training is carried out on Ubuntu 22.04 operating system equipped with 2 Nvidia 3090 GPU cards, PyTorch 2.0, CUDA 12.1 e TensorRT 9.3.0.1. A 100-iteration optimization process is then started with the collected dataset. The best training obtains a mAP@.50-95 = 0.85 ± 0.01, with initial learning rate of 0.000251, AdamW optimizer, batch size of 16, YOLOv8x-seg model, and trained to epoch 247 (i.e., interrupted by early stopping logic). The obtained model, once exported to TensorRT has an inference runtime of 6 ms (mean out of 1000 inferences). The model is then applied on the entire dataset collected (i.e., 457,000 images), and for each inference any information about the output mask and classes detected are stored.

### 2.3. Output Data Organization

The model, as described in Section 2.2, when applied to the entire dataset, returns information related to detected objects (i.e., AD) such as their location (i.e., centroid of the mask associated to detected AD), class, and the confidence percentage associated with that class. This collection of information is stored on a 2D matrix that then contains information for each of the 457,000 images, including the class of the object, the probability that the object belongs to that class, and the x- and y-coordinates of the object’s centroid (Table 1).

The first information is the object class (i.e., 0 for male, 1 for female and Not a Number, NaN if the object is not present in the frame). The second one is the probability that the object actually belongs to that class, this number is defined by the average of the associated probabilities belonging to the identified object. The third and fourth data are the object centroid coordinates which allow to determine the position of the AD within the frame and therefore along the bridge connecting BOX1 to BOX2. These information are important because it is not possible to base the count on the actual ADs identified in all the images. In fact the same AD is moving across the bridge (i.e., the transparent tube) from left to right and therefore in sequential images we will always found the same AD during its travel from BOX1 to BOX2. This AD must be counted only once. We could expect then to have a sequence of images with the AD center moving from left to right, and afterwords, a sequence with no AD if any other insect enters into the bridge, or another sequence with the AD center moving from left to right. But it can happen that an AD comes back towards BOX1. This situation can be filtered out by calculating the travel direction and keeping only the left to right one. Another circumstance is that in one frame more than one AD can exist because few ADs can travel together from BOX1 to BOX2. And therefore the 2D output matrix described before will be replicated in a third dimension where the same data will be stored but for the second object identified. In conclusion the output matrix will be a 3D table with the four information described in Table 1 repeated in the third dimension for each AD found in the same image (see the scheme in Figure 3).

## 3. Post-Process Methodology for Gender-Based Counting of *Acheta domesticus*

The complete flow chart of the counting procedure is summarised in Figure 4, including the different phases:image acquisition;AI-based detection;AD counting process;AD gender counting process.

**Figure 4 sensors-24-04936-f004:**
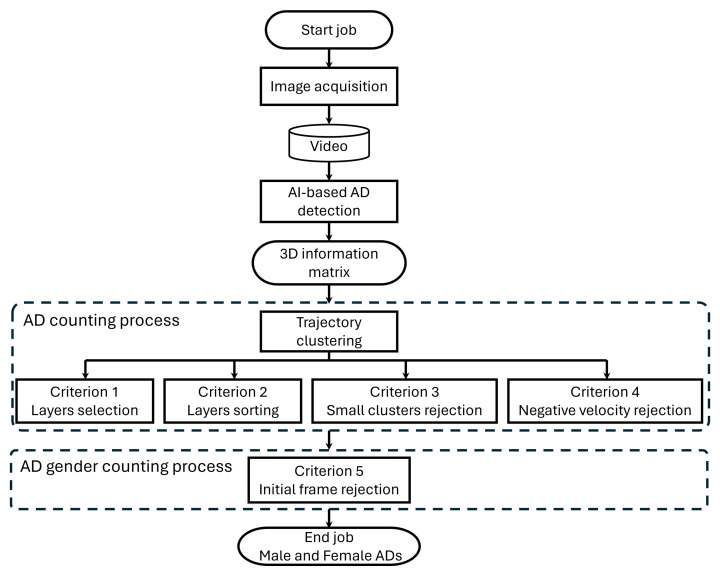
Work flow of the gender-based counting of *Acheta domesticus*.

### 3.1. AD Identification and Tracking

The acquisition resulted in 457,000 frames, but the YOLOv8-based semantic segmentation model identifies the presence of ADs only in 42,944 frames. Among them, two individuals were detected in the same image in 2038 frames, three individuals were detected in the same image in 65 frames, and four individuals were detected in the same image in 5 frames. In the multiple recognition frames, each AD is allocated in the third dimension of the information matrix (see Figure 3) which can be considered as a layered 2D matrix. It can be seen that the first and second layers contain the majority of ADs, while the third and fourth layers together account for only 0.15% of the total. Due to this observation, it was decided to consider only the first and second layers thus neglecting frames in which no more than two crickets were detected. This resulted in forcing the 3D information matrix to have the third dimension equal to 2. This suggests that the transfer channel between the two reservoirs has a fairly satisfactory conformation that avoids overlapping of the various ADs on each other, even if the size of the duct is reduced a more orderly flow of crickets would occur. The layer selection operation is the first criterion (Criterion 1) used in the counting process whose flow chart is described in Figure 4.

The results obtained were processed primarily for the purpose of ordering the individuals recognized on the various layers through a labeling process. In fact, in the case of multiple recognition’s the same individual may be attributed to the first layer in one frame and to the second layer in the next frame. Should this occur, discontinuities would be created in the trajectory of a single AD with detrimental effects on subsequent operations. Therefore, a sorting algorithm that performs individuals labeling by imposing a condition of continuity in their trajectory, swapping between two layers, was developed. In Figure 4, this processing step is called Criterion 2. It is based on the calculation of velocity ($v_{ij}$) and acceleration ($a_{ij}$) of each individual following Equations (Equation 2) and (Equation 3) where i,j∈{Layer1,Layer2}; dt is defined as the reciprocal of the acquisition frame rate, *f* is the actual frame, $f^{\prime \prime }$ is the frame following that one/those ones where the target body part $p_{n}$ is missing (i.e., the frame following a series of one or more NaNs) and $f^{\prime }$ is the frame preceding that one/those ones where the target body part *n* is missing. The optimized trajectory is the one that minimizes acceleration and velocity, eliminating the jumps that occur when the same AD mistakenly is not attributed to the same layer between frames.
(2)$$\begin{array}{c} \left\{\begin{array}{c} v_{ii}\left(f\right)=\frac{p_{n,i}\left(f\right)-p_{n,i}\left(f-1\right)}{dt}\\ v_{ij}\left(f\right)=\frac{p_{n,j}\left(f\right)-p_{n,i}\left(f-1\right)}{dt} \end{array}\right. \end{array}$$
(3)$$\begin{array}{c} \left\{\begin{array}{c} a_{ii}\left(f\right)=\frac{v_{i}\left(f^{\prime \prime }\right)-v_{i}\left(f^{\prime }\right)}{t\left(f^{\prime \prime }\right)-t\left(f^{\prime }\right)}\\ a_{ij}\left(f\right)=\frac{v_{j}\left(f^{\prime \prime }\right)-v_{i}\left(f^{\prime }\right)}{t\left(f^{\prime \prime }\right)-t\left(f^{\prime }\right)} \end{array}\right. \end{array}$$

The coordinates of the first layer were switched with the coordinates of the second layer according to the minimization of velocity and acceleration between subsequent frames (e.g., if $v_{ij}$ and $v_{ji}$ were lower then $v_{ii}$ and $v_{jj}$). In Figure 5, it is possible to observe an example of wrists trajectories before and after the application of the sorting algorithm. For the sake of simplicity it is shown only the x-coordinate of the two ADs framed in the images sequence considered. At the beginning only one AD is present in the frame and it will be attributed to first layer of the information matrix. Its trajectory is marked with blue stars in Figure 5a. At frame 36, a second AD appears in the scene which is attributed to the second layer whose trajectory is marked with red stars. It can be noticed that at frame 39, a layer swap occurs. In fact, the first AD is now attributed to the second layer and its trajectory becomes red. The same happens for the second AD, which now is attributed to the first layer and its trajectory modifies in blue. By applying the sorting algorithm, the inconsistency in mixing the assignment of layers is solved (see Figure 5b), thus making it possible to perform further analyses.

Once identified the trajectory of the first AD and of the second one, if the second layer is not empty, the counting of the individuals can be updated by performing a clustering operation on the trajectories based on a differentiation process so as to separate each AD passage within the duct from the next one. In order to minimize the influence of measurement noise, some trajectories have been discarded in the counting operation, using Criterion 3, shown in Figure 4. In particular, trajectories consisting of a small number of points, probably due to noisy or otherwise isolated images, were discarded. In order to identify the minimum cluster size, a statistical analysis of the speed at which the ADs move across the duct was performed. The statistical distribution of the ADs speed is shown in Figure 6, for both the two layers. The distribution appears fairly Gaussian with a slight Skewness favoring the highest speed. The average value of the velocity is 50 px per frame which corresponds to about 13 mm/s, being the frame rate 5 px/s and the pixel size 48.1 μm. This velocity value is also confirmed in the state-of-the-art based on [36], where Storm et. al. reported the speed of movement of insects. The analysis of the ADs speed suggests that clusters with fewer than 15 frames identified correspond to clusters in which ADs would be expected to move with an improbable speed, i.e., speed higher than 120 px/frame (corresponding to the 95 percentile of the distribution reported in Figure 6), and thus can be discarded. Figure 6 reveals also that the probability of having two ADs moving within the scene framed by the camera is small (only about 92 compared to the 1189 counted on the first layer). This fact demonstrates that the experimental setup was correctly designed to allow only one AD at a time to enter the duct.

Another criterion for selecting ADs, Criterion 4 in Figure 4, is their movement direction. In fact, an AD may move backwards. Therefore, in this case it must be discarded because it did not move to BOX2 but returned to BOX1. There are also cases where within the same cluster, the ADs linger moving back and forth.

### 3.2. AD Gender Identification

Once the trajectories of each AD passing across the duct have been identified, the gender recognition must be performed, taking into account the information about the class of the individuals retrieved by the semantic segmentation model and stored in the first column of the matrix, as described in Table 1 and reported in Figure 3. As explained in Section 1.1, the recognition of females is mainly determined by the presence of the ovipositor, but this is subject to errors due to two main reasons. The first reason is linked to the possibility that the ovipositor is hidden or simply confused with one of the AD legs. This is mainly a random error component. The second reason, which can be considered a systematic error component, is related to the fact that the ovipositor, being located in the back of the AD, is not in the camera’s field of view in the first frames in which the AD enters into the duct. In this case, the segmentation model detects AD but, because the back is not framed in the image, it may misclassify AD. This fact suggests discarding the first frames of each cluster, in which systematically all ADs will be recognized as males, following Criterion 5 described in Figure 4.

## 4. Discussion of Results and Uncertainty Analysis

Before presenting the final result on the gender-based ADs counting, a synthesis of the counting procedure and uncertainty sources that influence it is presented by considering a set of frames described in Figure 7. Figure 7 reports the trajectory of the identified ADs in the x-direction (only this direction is considered for the sake of simplicity in visualisation) from frame number 101,000 to frame number 101,100. It must also be pointed out that Figure 7 represents the individuals information stored in the first layer of the 3D information matrix described in Figure 3. Therefore, if two ADs are present in one frame, the information related to the second one will be stored in the second layer and it will not be shown here. If the AD class is male, the marker in the trajectory is colored in green, if it is female, it is colored in red. Some images captured at different positions are superimposed to the trajectory plot. It can be observed that the frame 101,026 contains a female AD (evidenced by the ovipositor in its back) which is correctly recognised. At frame 101,034, a second male individual enters into the scene which is correctly recognised but is wrongly attributed to the first layer. This can be considered an uncertainty because the AD cannot suddenly change their gender. It is also clear that this is an error because the individual is considered as male only for seven non-consecutive positions and then it is recognised again as female. This uncertainty can be solved by taking into account the AD velocity and acceleration data and also considering the two layers of the 3D information matrix in conjunction (Criterion 2). At frame 101,063 the first AD (the female one) has left the duct and only the male one is framed. Therefore this individual is now attributed to the first layer. Another source of uncertainty that occurs at frame 101070 is the wrong classification due to the failing of the AI model in correctly identifying the ovipositor. In fact, in this case the rear tergites have been confused with the ovipositor and the individual has been wrongly classified as female. Finally, at frame 101,075, another individual enters the scene (this frame is not reported as an image in Figure 7) and another cluster trajectory is identified. If we observe the AD shot at the 101,089 frame, we clearly recognise the ovipositor, which was not identified by the AI model in the first 13 frames which are green. This error represents the systematic error due to the fact that in the entering of the AD into the scene, its back is not framed and therefore the algorithm assigns it to the male class.

### 4.1. Discussion of Results

In this section, the counting result will be discussed by considering the different criteria, from 1 to 5 as summarised in Figure 4, used to improve the accuracy in the AD number calculation. The first criterion is the one used to consider the presence in the same image of two or more ADs which accounts of different layers of the 3D information matrix. If only the first layer is considered, the total number of ADs is 1189. Consequently, we would register an underestimation of the 7% with respect to the number of ADs counted manually (1281 insects). The second criterion is the one based on the sorting algorithm imposing the continuity condition in the ADs trajectory, as visualised in Figure 5. If the sorting algorithm is not applied, the total number of ADs is 1346 resulting in an overestimation of 5.1%. The third criterion is the discarding of ADs whose trajectory contains a small number of frames, which testifies a high speed in crossing the bridge. As explained in Section 3, a cluster containing less than 15 elements should be discarded. This is demonstrated by observing Figure 8, showing how the correct choice of cluster size impacts on the counting error. It can be noticed that including small clusters (with a number of individuals found less than 15) leads to the count being overestimated. Discarding clusters containing more than 15 individuals results in the count being underestimated.

Finally, the fourth criterion for the ADs counting is the accounting for the AD moving direction which must be from BOX1 to BOX2. If the trajectories with negative sign, corresponding to ADs moving in the opposite direction (from BOX2 to BOX1), are considered, an overestimation will occur: the number of ADs counted is 1250 which corresponds to an error of 2.4%. However, this error is almost negligible, testifying that the inclusion on the ADs selection of the direction (or better the sign) of the velocity, to compensate the aforementioned anomalies, had a very little impact on the counting accuracy. This result can prove the fact that the duct design favors a fairly uniform and smooth flow of ADs. When all the previous criteria are implemented, the total number of ADs is 1247 which corresponds to an underestimation in the counting of 2.6 %. This error is higher than the one we have when the direction of the AD passage is not considered (2.4 %), but this criterion can not be neglected because the double passage of ADs that travel from BOX1 to BOX2 and return must be excluded.

The last criterion (Criterion 5 in Figure 4) to be adopted is related to gender classification. In Section 3.2, we described that the first frames where the AD appears in the scene must be discarded because the ovipositor is not visible and thus the AD will all be classified as male. Here, again the speed calculation comes to our aid and suggests discarding the first three frames in the gender computation. The effect of the number of discarded initial frames on the male/female counting error is reported in Figure 9. It can be observed that the minimum error is obtained if the first three frames are rejected.

Finally, if all the criteria are applied the number of females is 683 and the number of males is 564. Considering that the females and males measured with the manual counting were 747 and 534, respectively, we have an error of −8.6% and 5.6% in the gender classification. The negative error states that the counting process underestimates the females, while the males are overestimated. The estimated error in counting the ADs and their gender is summarized in Table 2 which gives a clear indication of the improvement provided by the application of the selection criteria described in Section 3.1 and reported in Figure 4. The positive sign of the counting error indicates an overestimation, the negative sign an underestimation.

### 4.2. Uncertainty Analysis

According to [37], in deep learning data processing there are two major different types of uncertainty: the epistemic and the aleatory uncertainties. The epistemic uncertainty describes what the model does not know because training data were not appropriate, due to limited data and knowledge, which affects the capability of training to generate a robust model. High epistemic uncertainty arises when data are incomplete, for instance, where there are few or no observations that can be used in the training process. The aleatory uncertainty is related to stochasticity of observations. It can be reduced not by increasing the quantity of data provided, but by improving the quality of data. This is the typical uncertainty defined in measurement science. Common sources of aleatory uncertainty are, for example, sensor malfunction, inadequate illumination or optical window quality (the perspex transparency of the bridge between BOX1 and BOX2). These sources influence what can be considered as measurement noise.

In the case study we present in this paper, we can distinguish three aspects which affect the global uncertainty both from an epistemic and aleatory point of view:identification of the AD;its location within the image;the recognition of the gender of the AD.

With regard to the identification of the AD, it is based on the definition of a score (mAP) and a threshold. The uncertainty aspect then resides on the performance of the score in the various frames (how abruptly the score varies between frames with and without AD) and the identification of the correct threshold. This can lead to false positives or false negatives.

As far as the location of the AD in the frame is concerned, the uncertainty is related to the pixel size and in this case it is very low because of the high resolution of the camera used. This contribution can be considered random. In fact it is conditioned by the general quality of the image, as cleanliness of the window, level of illumination and its distribution, position assumed by the body of the AD in each frame. The location of the AD is used in the counting process for the identification of the individual trajectory which will be exploited in the clustering process for the AD individuation. Typical trajectories are shown in Figure 5 and Figure 7, evidencing a smooth trend, this meaning that the positioning uncertainty in a single frame does not evidently affect the global trend of the trajectory. Therefore, this uncertainty does not have an important impact on the clustering used for the counting process.

Finally, the uncertainty related to the recognition of the AD gender is linked to the identification of the ovipositor. Basically, the problem arises when the ovipositor turns out to be hidden from view, thus leading to nonrecognition in the female case and overestimating the male population or when the network mistakenly recognizes a limb as an ovipositor element, in this case overestimating the female population. From the experiments performed, we can see that it is the former case that occurs most frequently and we can therefore conclude a tendency for the network to overestimate the male population. However, this observation made it possible to partially correct this effect in post-processing. Essentially, if in some frames of a trajectory the individual is recognized as male and in some other frames as female, the entire trajectory can be reasonably attributed to a female specimen under the assumption that a simple occlusion of the ovipositor occurred.

## 5. Conclusions

This paper presents a critical analysis of a vision-based measurement system for the automatic counting of male and female AD in cricket farms. Such a system can have a strong impact on the process optimization in insect mass rearing since the male–female ratio balance influences quality and quantity of insects produced. Another benefit is related to the food industry given the nutritional characteristics related to the gender. An identical cricket farm is proposed for use where the entire cricket colony can be transferred. The two farms are connected by a transparent tube, allowing the crickets to spontaneously move from one farm to the other. The new farm is made more attractive by modulating factors such as lighting, water, food, and temperature. The methodology involves capturing images of ADs as they pass through the Perspex tube connecting the two boxes, using a high-resolution camera. This setup collects a series of images that are then processed through a Deep Learning-based model capable of recognizing the location of ADs within the frame and identifying their gender. The data collected are then processed by a counting algorithm implementing several criteria to mitigate the effect of uncertainty sources, like:the presence of more than one AD in the same frame,the AD trajectory discontinuities due to AD mismatching,the identification of trajectories with a small number of frames, corresponding to ADs moving too fast,the eventuality of ADs that come back and have a negative velocity.

The algorithm also implements a criterion to address another source of uncertainty that affects gender counting. Specifically, in the initial frames of the AD trajectory, the ovipositor is hidden, causing the segmentation model to incorrectly attribute a male gender to those frames. By considering all those corrections, the error in the ADs number estimation is reduced from 7 to 2.6%. Considering the gender counting error, this estimation is reduced from −10.5 to −8.6%, for females, and from 8.3 to 5.6%, for males, if the first frames of the trajectory are kept or discarded, respectively. It has been evidenced that there is always an underestimation of females (negative error) and an overestimation of males (positive error) due to the fact that, even if all the correction criteria are applied, the presence of the ovipositor can be unidentified because of noise and other random sources. However, the level of those errors can be considered satisfactorily low for the application case considered.

## Figures and Tables

**Figure 1 sensors-24-04936-f001:**
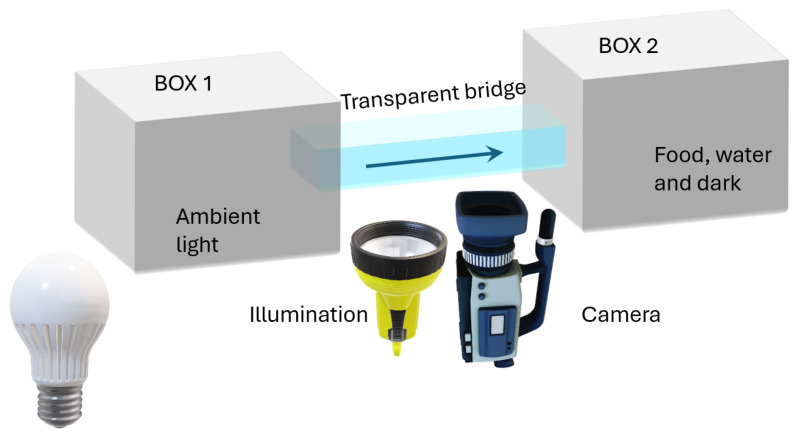
Schematic of the proposed test bench for the proposed gender-based counting of *Acheta domesticus*.

**Figure 2 sensors-24-04936-f002:**
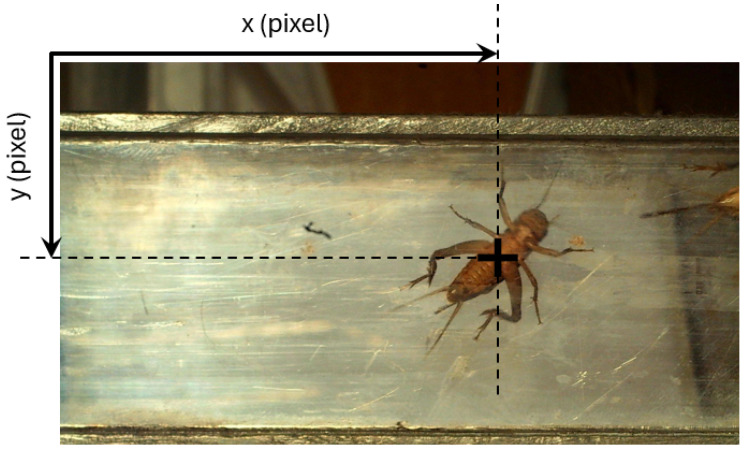
Example of the x- and y-coordinates of the AD’s centroid in the image.

**Figure 3 sensors-24-04936-f003:**
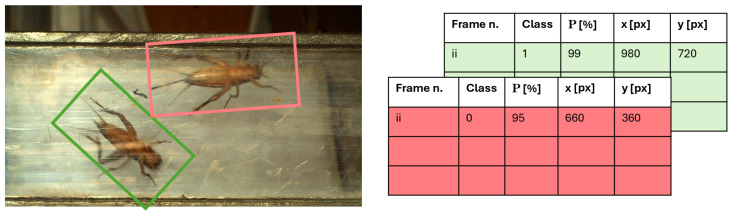
Scheme of the output data from the AI model which are organised as the 3D matrix sketched in the left side. The matrix dimension is l × m × n, with l n. of frames, m n. of information (4), n n. of ADs in the frame or n. of layers (2 in this case).

**Figure 5 sensors-24-04936-f005:**
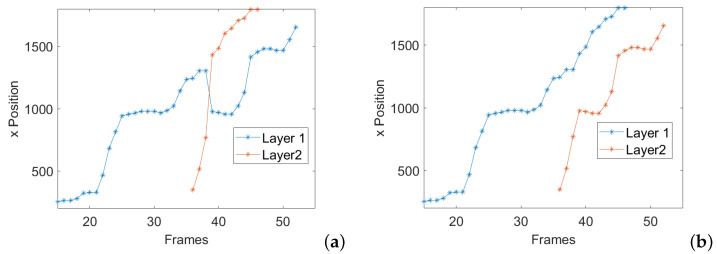
Example of sorting algorithm: Before sorting (**a**), after sorting (**b**).

**Figure 6 sensors-24-04936-f006:**
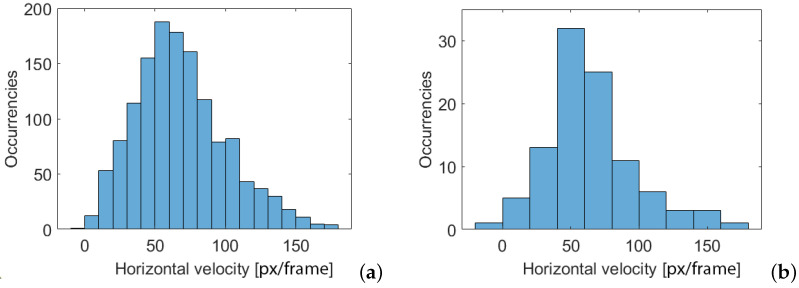
Histogram of ADs velocity expressed in [px/s]: Layer 1 (**a**) and Layer 2 (**b**).

**Figure 7 sensors-24-04936-f007:**
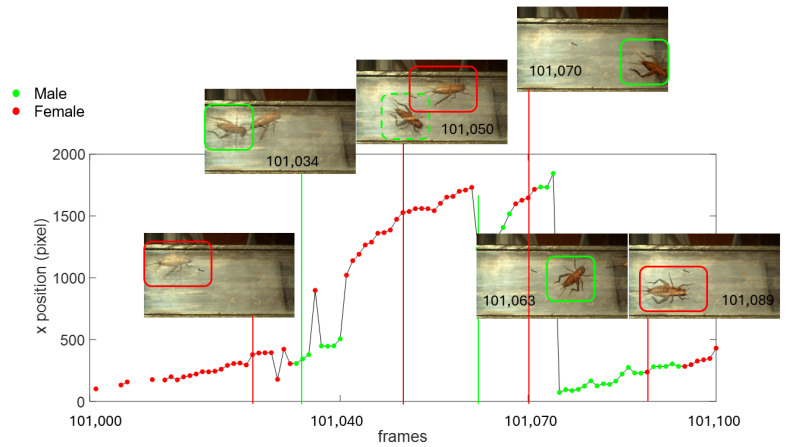
ADs trajectories and gender detection.

**Figure 8 sensors-24-04936-f008:**
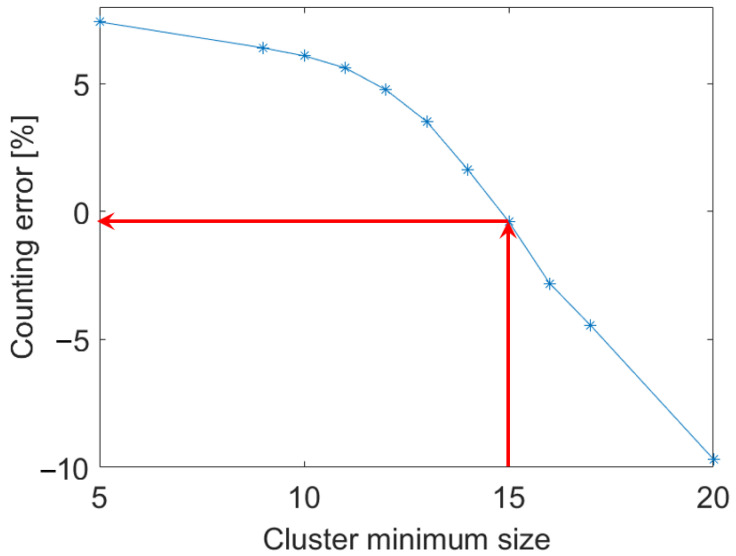
Impact of cluster minimum size on ADs counting error.

**Figure 9 sensors-24-04936-f009:**
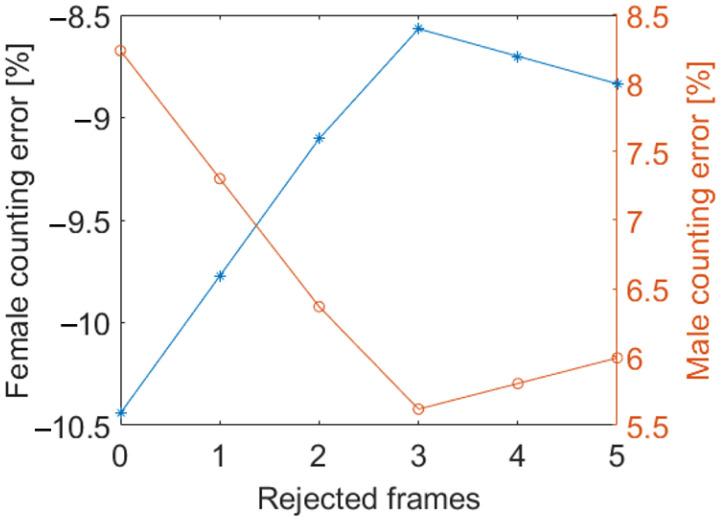
Impact of number of initial frames discarded on ADs male/female counting error.

**Table 1 sensors-24-04936-t001:** The model, when applied to the entire dataset, returns information related to detected objects (i.e., AD) such as their location (i.e., centroid of the mask associated to detected AD), class, and the confidence percentage associated with that class.

Information	Description [Measurement Unit]	Comment
Object class	ADs gender	0 if male 1 if female NaN (Not a Number) if no AD
Probability that		
the object actually belongs	[%]	
to that class		
	x-coordinate of the center	
Centroid x-coordinate	of the object identified	See Figure 2
	in the frame [px]	
	y-coordinate of the center	
Centroid y-coordinate	of the object identified	See Figure 2
	in the frame [px]	

**Table 2 sensors-24-04936-t002:** Summary of the counting error if Criteria from 1 to 5 are/are not applied.

Criterion # and Description		Maximum Counting Error [%]	Residual Error [%]
1. Layer selection		−7.0	+2.6
2. Layer sorting		+5.1	
3. Small cluster rejection		−10.0	
4. Negative velocity rejection		+2.4	
5. Initial frames rejection	Male	+8.4	+5.6
	Female	−10.5	−8.6

## Data Availability

Data are contained within the article.

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
