# Peer review of "Vision Measurement System for Gender-Based Counting of Acheta domesticus"

_sensors, 2024, doi:10.3390/s24154936_

Round 1

Reviewer 1 Report

Comments and Suggestions for Authors

The article presents a non-contact measurement system for sex counting in house crickets, which has potential applications in insect farm management, especially where sex ratio has a significant impact on reproduction and production.

1. the introduction section needs to discuss in more detail the limitations of the prior art and the innovations of this study.

2. in Dataset pre-processing and model training and Hyper-parameter Optimization Process I think a serial number should be added.

3. line 120? Please note these details

4. the sequential position of this figure is very inconvenient to read suggest to adjust the order.

5. Could Figure 5 perhaps reveal more specific details?

6. The conclusions of the results need to be presented in clearer graphs and images. More details about the validity of the model performance evaluation should be provided. 7.

7. the discussion section should provide an in-depth analysis of the significance of the results, including comparisons with existing literature, limitations of the methodology and possible directions for improvement.

8. the conclusion section should more clearly summarize the main findings of the study and indicate directions for future research.

9. A more comprehensive literature review may be required.

Author Response

Comments: 

The article presents a non-contact measurement system for sex counting in house crickets, which has potential applications in insect farm management, especially where sex ratio has a significant impact on reproduction and production.

1. the introduction section needs to discuss in more detail the limitations of the prior art and the innovations of this study.

2. in Dataset pre-processing and model training and Hyper-parameter Optimization Process I think a serial number should be added.

3. line 120? Please note these details

4. the sequential position of this figure is very inconvenient to read suggest to adjust the order.

5. Could Figure 5 perhaps reveal more specific details?

6. The conclusions of the results need to be presented in clearer graphs and images. More details about the validity of the model performance evaluation should be provided. 7.

7. the discussion section should provide an in-depth analysis of the significance of the results, including comparisons with existing literature, limitations of the methodology and possible directions for improvement.

8. the conclusion section should more clearly summarize the main findings of the study and indicate directions for future research.

9. A more comprehensive literature review may be required.

Responses:

The authors would like to thank the reviewer for the valuable feedback and appreciation of their work.
We have proceeded to rewrite, for the most part, the introductory part of the paper. We have added more subsections, added references, and highlighted more clearly and comprehensively the motivations and innovative part of the proposed work.
We have corrected incorrectly linked citations and missing paragraph numbers.
Another detail that is revealed by Figure 5, that now is Figure 6, is that the probability of two ADs moving within the scene framed by the camera is small (only about 92 compared to the 1189 counted on the first layer). This shows that the experimental setup was correctly designed to allow only one AD at a time to enter the duct.
We have also rearranged the pictures so that they appear in the right section to make reading more fluent. The discussion and conclusion parts have been expanded, and a summary table has also been added to make the performance of the method proposed in this paper clearer.
All changes made are shown in red text.

Reviewer 2 Report

Comments and Suggestions for Authors

This paper addresses an interesting topic for insect mass rearing: a technique for the identification and consequently optimization of gender separation in crickets.

Some specific corrections are necessary:

- The binomials related to the scientific names of insects need to be highlighted in italics.

- on lines 59-60: six species: which?

- line 66: its capabilities;

- line 75: “natural” environment?

- line 120: implemented in [?]?

- line 130: “is a common metric”: reference(s)?

- line 169: if any other insect enters...?

- line 176: 4 informations;

- line 177: (see the scheme in Figure 3);

- line 194: why Figure 6 appears before the Figure 4?

- line 243: “movement direction” intead of “movement speed direction”?

- line 251: “As explained in Section 2.2” or in Section 1.1 (lines 82-823)?

- line 293: frame 101075 does not appear in the Figure 7;

- line 318: underestimation? 1250 in relation to 1281counted manually;

- line 319: the inclusion of direction on the ADs selection to compensate...

- Figures 8 and 9 cited in the paragraphs of the Discussion. Shouldn't they appear earlier in the Results?

- line 352: influence what can be considered...

- lines 2 and 387: wouldn’t it be better to replace “sustainability of farming” by “process optimization in insect mass rearing”?

- line 501: towardsdatascience.com

Comments on the Quality of English Language

Minor corrections are necessary.

Author Response

Comments:

This paper addresses an interesting topic for insect mass rearing: a technique for the identification and consequently optimization of gender separation in crickets. Some specific corrections are necessary.

Responses:

The authors would like to thank you for the feedback received and appreciation of our work. 
We corrected all reported errors, moved figures and rewrote much of the introduction and commentary to the results, then the conclusions. Corrected several errors in grammar and verb forms. All corrections are highlighted in the text in red color.
We remain available for any further questions or requests.